# Bayesian Pseudocoresets

**Dionysis Manousakas**
Department of Computer Science & Technology
University of Cambridge
dm754@cam.ac.uk

**Zuheng Xu**
Department of Statistics
University of British Columbia
zuheng.xu@stat.ubc.ca

**Cecilia Mascolo**
Department of Computer Science & Technology
University of Cambridge
cm542@cam.ac.uk

**Trevor Campbell**
Department of Statistics
University of British Columbia
trevor@stat.ubc.ca

## Abstract

Standard Bayesian inference algorithms are prohibitively expensive in the regime of modern large-scale data. Recent work has found that a small, weighted subset of data (a *coreset*) may be used in place of the full dataset during inference, taking advantage of data redundancy to reduce computational cost. However, this approach has limitations in the increasingly common setting of sensitive, high-dimensional data. Indeed, we prove that there are situations in which the Kullback-Leibler (KL) divergence between the *optimal* coreset and the true posterior grows with data dimension; and as coresets include a subset of the original data, they cannot be constructed in a manner that preserves individual privacy. We address both of these issues with a single unified solution, *Bayesian pseudocoresets*—a small weighted collection of synthetic "pseudodata"—along with a variational optimization method to select both pseudodata and weights. The use of pseudodata (as opposed to the original datapoints) enables both the summarization of high-dimensional data and the differentially private summarization of sensitive data. Real and synthetic experiments on high-dimensional data demonstrate that Bayesian pseudocoresets achieve significant improvements in posterior approximation error compared to traditional coresets, and that pseudocoresets provide privacy without a significant loss in approximation quality.

## 1 Introduction

Large-scale data—which has become the norm in many scientific and commercial applications of statistical machine learning—creates an inherently difficult setting for the modern data analyst. Exploring such data is difficult because it cannot all be obtained and directly visualized at once; one is typically limited to accessing potentially nonrepresentative random subsets of data. Exploring models is similarly hard, as training even a single model can be a computationally expensive, slow, and unreliable process. And as many sources of large-scale data contain sensitive information about individuals (e.g., electronic health records and social network data), these challenges are coupled with growing privacy concerns that preclude direct access to individual datapoints completely.

Large-scale data does offer one reprieve to the analyst: it often exhibits a significant degree of redundancy. Most data are not unique or particularly informative for modelling and exploration. Based on this notion, data summarization methods have been developed that provide the practitioner with a compressed—but still statistically representative—version of the large dataset for analysis. Summarizations have been developed for a variety of purposes, e.g., reducing the cost of computing with kernel matrices via Nyström-type approximations [11, 32, 3] or sparse pseudo-input parameterizations [37], Bayesian inference [23, 22, 8, 9, 7], maximum likelihood parameter estimation

[12, 30], linear regression [43, 20], geometric shape approximation [2], clustering [16, 29, 4, 6], and dimensionality reduction [18].

A common form of summarization is that of a sparse, weighted subset of the original dataset—a *coreset* [2]. Coresets have two distinct advantages over other possible summarization modalities: they are easily interpreted, and can often be used as the input to standard data analysis algorithms without modification. But as the dimensionality of a dataset grows, its constituent datapoints tend to become more "unique" and cannot represent one another well. Indeed, in the context of Bayesian inference—the focus of the present work—we show that the *optimal* coreset posterior approximation to the true posterior has KL divergence that scales with the dimension of the data in a simple problem setting (Proposition 1). Furthermore, directly releasing a subset of the original data precludes any possibility of individual privacy under the current standard of differential privacy [14, 15]. Past work addresses this issue in the context of clustering and computational geometry [17, 19], while the idea of releasing private dataset compressions has been also pursued in kernel mean embeddings [5], sparse regression [43], and compressive learning [36]—with the remarkable property that the privatized compressed data may be queried *ad infinitum* without loss of privacy—but no such method exists for Bayesian posterior inference.

In this work, we develop a novel technique for data summarization in the context of Bayesian inference under the constraints that the method is scalable and easy to use, creates an intuitive summarization, applies to high-dimensional data, and enables privacy control. Inspired by past work [30, 43, 37], instead of using constituent datapoints, we use synthetic *pseudodata* to summarize the large dataset, resulting in a *pseudocoreset*. We show that in the high-dimensional problem setting of Proposition 1, the optimal pseudocoreset with just one pseudodata point recovers the exact posterior, a significant improvement upon the optimal standard coreset of any size. As in past work on Bayesian coresets [7], we formulate pseudocoreset construction as variational inference, and provide a stochastic optimization method. As a consequence of the use of pseudodata—as well as privacy-preserving stochastic gradient descent mechanisms [1, 34, 27]—we show that our method can easily be modified to output a privatized pseudocoreset. The paper concludes with experimental results demonstrating the performance of pseudocoresets on real and synthetic data.

## 2 Bayesian Coresets

In this work, the goal is to approximate expectations under a density $\pi(\theta)$, $\theta \in \Theta$ expressed as the product of $N$ potentials $(f(x_n, \theta))_{n=1}^N$ and a base density $\pi_0(\theta)$:

$$\pi(\theta) := \frac{1}{Z} \exp \left( \sum_{n=1}^N f(x_n, \theta) \right) \pi_0(\theta).$$

In the setting of Bayesian inference with conditionally independent data, the potentials are data log-likelihoods, i.e. $f(x_n, \theta) := \log \pi(x_n|\theta)$, $\pi_0$ is the prior density, $\pi$ is the posterior, and $Z$ is the marginal likelihood of the data. Rather than working directly with $\pi(\theta)$ for posterior inference—which requires a $\Theta(N)$ computation per evaluation—a Bayesian coreset approximation of the form

$$\pi_w(\theta) := \frac{1}{Z(w)} \exp \left( \sum_{n=1}^N w_n f(x_n, \theta) \right) \pi_0(\theta)$$

for $w \in \mathbb{R}^N$, $w \geq 0$ may be used in most popular posterior inference schemes [33, 28, 35]. If the number of nonzero entries $\|w\|_0$ of $w$ is small, this results in a significant reduction in computational burden. Recent work has formulated the problem of constructing a Bayesian coreset of size $M \in \mathbb{N}$ as sparse variational inference [7],

$$w^\star = \underset{w \in \mathbb{R}^N}{\arg \min} \, D_{KL} \left( \pi_w || \pi \right) \qquad \text{s.t.} \quad w \geq 0, \ \|w\|_0 \leq M, \tag{1}$$

and showed that the objective can be minimized using stochastic estimates of $\nabla_w D_{KL} \left( \pi_w || \pi \right)$ based on samples from the coreset posterior $\pi_w$.

### 2.1 High-dimensional data

Coresets, as formulated in Eq. (1), are limited to using the original datapoints themselves to summarize the whole dataset. Proposition 1 shows that this is problematic when summarizing high-dimensional data; in the common setting of posterior inference for a Gaussian mean, the KL divergence $D_{KL} \left( \pi_{w^\star} || \pi \right)$ of the *optimal* coreset of any size scales with the dimension of the data. The proof may be found in Supp. A.

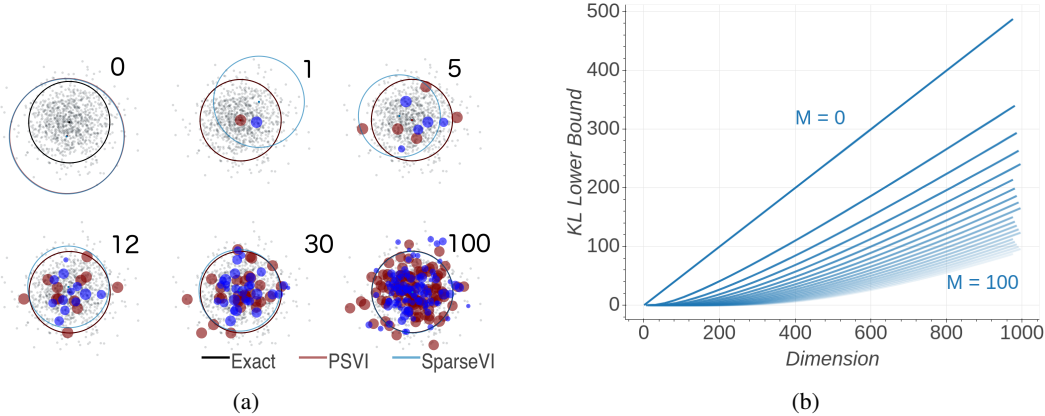

Figure 1: Gaussian mean inference under pseudocoreset (PSVI) against standard coreset (SparseVI) summarization for $N = 1,000$ datapoints. (a) Progression of PSVI vs. SparseVI construction for coreset sizes $M = 0, 1, 5, 12, 30, 100$, in 500 dimensions (displayed are datapoint projections on 2 random dimensions). PSVI and SparseVI coreset predictive $3\sigma$ ellipses are displayed in red and blue respectively, while the true posterior $3\sigma$ ellipse is shown in black. PSVI has the ability to immediately move pseudopoints towards the true posterior mean, while SparseVI has to add a larger number of existing points in order to obtain a good posterior approximation. See Fig. 2b for the quantitative KL comparison. (b) Optimal coreset KL divergence lower bound from Proposition 1 as a function of dimension with $\delta = 0.5$, and coreset size $M$ evenly spaced from 0 to 100 in increments of 5.

**Proposition 1.** *Suppose we use $(X_n)_{n=1}^N \overset{i.i.d.}{\sim} \mathcal{N}(0, I)$ in $\mathbb{R}^d$ to perform posterior inference in a Bayesian model with prior $\mu \sim \mathcal{N}(0, I)$ and likelihood $(X_n)_{n=1}^N \overset{i.i.d.}{\sim} \mathcal{N}(\mu, I)$. Then $\forall M < d$ and $\delta \in [0, 1]$, with probability at least $1 - \delta$ the optimal size-$M$ coreset $w^\star$ satisfies*

$$D_{\mathrm{KL}}\left(\pi_{w^\star}||\pi\right) \geq \frac{1}{2} \frac{N - M}{1 + N} F_{d-M}^{-1}\left(\delta \binom{N}{M}^{-1}\right),$$

*where $F_k$ is the CDF of a $\chi^2$ random variable with $k$ degrees of freedom.*

The bound in Proposition 1 depends on $d$ through the $\chi^2$ distribution inverse CDF. Although difficult to see directly, the bound is reasonably large for typical values of $N, M, d, \delta$, and increasing linearly in $d$; Fig. 1b visualizes the value of the lower bound as a function of dimension $d$ for various coreset sizes $M$. Note that the above bound requires the data to be high-dimensional such that $d > M$; if $d \leq M$ the proof technique in Supp. A results in a vacuous $D_{\mathrm{KL}}\left(\pi_{w^\star}||\pi\right) = 0$ lower bound.

## 3 Bayesian Pseudocoresets

Proposition 1 shows that there is room for improvement in coreset construction in the high-dimensional data regime. Indeed, consider again the same problem setting; the coreset posterior distribution is a Gaussian with mean $\mu_w$ and covariance $\Sigma_w$,

$$\Sigma_w = \left(1 + \sum_{n=1}^N w_n\right)^{-1} I \qquad\qquad \mu_w = \Sigma_w \sum_{n=1}^N w_n X_n. \qquad (2)$$

Examining Eq. (2), we can replicate any coreset posterior exactly by using a single synthetic *pseudodata* point $U = \left(\sum_{n=1}^N w_n\right)^{-1} \sum_{n=1}^N w_n X_n$ with weight $\sum_{n=1}^N w_n$. In particular, the true posterior is equivalent to the posterior conditioned on the single pseudodata point $U = \frac{1}{N}\sum_{n=1}^N X_n$ with weight $N$ (with corresponding KL divergence equal to 0). This is not surprising; the mean of the data is precisely a sufficient statistic for the data in this simple setting. However, it does illustrate that carefully-chosen pseudodata may be able to represent the overall dataset—as "approximate sufficient statistics"—far better than any reasonably small collection of the original data. This intuition has been used before, e.g., for scalable Gaussian process inference [37, 38], privacy-preserving compression in linear regression [43], herding [42, 10, 25], and deep generative models [39].

In this section, we extend the realm of applicability of pseudopoint compression methods to the general class of Bayesian posterior inference problems with conditionally independent data, resulting in *Bayesian pseudocoresets*. Building on recent work [7], we formulate pseudocoreset construction as a variational inference problem where both the weights and pseudopoint locations are parameters of the variational posterior approximation, and develop a stochastic algorithm to solve the optimization.

## 3.1 Pseudocoreset variational inference

A Bayesian pseudocoreset takes the form

$$\pi_{u,w}(\theta) = \frac{1}{Z(u,w)} \exp \left( \sum_{m=1}^{M} w_m f(u_m, \theta) \right) \pi_0(\theta),$$

where $u := (u_m)_{m=1}^{M}$ are $M$ pseudodata points $u_m \in \mathbb{R}^d$, $(w_m)_{m=1}^{M}$ are nonnegative weights, $f : \mathbb{R}^d \times \Theta \to \mathbb{R}$ is a potential function parametrized by a pseudodata point, and $Z(u,w)$ is the corresponding normalization constant rendering $\pi_{u,w}$ a probability density. In the setting of Bayesian posterior inference, $u_m$ will take the same form as the data, while the potentials are the log-likelihood functions, i.e., $f(u_m, \theta) = \log \pi(u_m|\theta)$. We construct a coreset by minimizing the KL divergence over both the pseudodata locations and weights,

$$u^\star, w^\star = \underset{u \in \mathbb{R}^{d \times M}, w \in \mathbb{R}_+^M}{\arg \min} \mathrm{D}_{\mathrm{KL}} \left( \pi_{u,w} || \pi \right). \tag{3}$$

As opposed to previous Bayesian coreset construction optimization problems [8, 9, 7], we do not need an explicit sparsity constraint; the coreset size is limited to $M$ directly through the selection of the number of pseudodata and weights.

Denote the vectors of original data potentials $f(\theta) \in \mathbb{R}^N$ and synthetic pseudodata potentials $\tilde{f}(\theta) \in \mathbb{R}^M$ as $f(\theta) := [f_1(\theta) \ldots f_N(\theta)]^T$ and $\tilde{f}(\theta) := [f(u_1, \theta) \ldots f(u_M, \theta)]^T$ respectively, where we suppress the $(\theta)$ for brevity where clear from context. Denote $\mathbb{E}_{u,w}$ and $\mathrm{Cov}_{u,w}$ to be the expectation and covariance operator for the pseudocoreset posterior $\pi_{u,w}$. Then we may write the KL divergence in Eq. (3) as

$$\mathrm{D}_{\mathrm{KL}} \left( \pi_{u,w} || \pi \right) = \mathbb{E}_{u,w}[\log \pi_{u,w}(\theta)] - \mathbb{E}_{u,w}[\log \pi(\theta)]$$
$$= \log Z(1) - \log Z(u,w) - 1^T \mathbb{E}_{u,w}[f] + w^T \mathbb{E}_{u,w}[\tilde{f}], \tag{4}$$

where $1 \in \mathbb{R}^N$ is the vector of all 1 entries, and $w \in \mathbb{R}^M$ is the vector of pseudocoreset weights.

As we will employ gradient descent steps as part of our algorithm to minimize the variational objective over the parameters $u, w$, we need to evaluate the derivative of the KL divergence Eq. (4). Despite the presence of the intractable normalization constants and expectations, we show in Supp. B that gradients can be expressed using moments of the pseudodata and original data potential vectors. In particular, the gradients of the KL divergence with respect to the weights $w$ and to a single pseudodata location $u_m$ are

$$\nabla_w \mathrm{D}_{\mathrm{KL}} = - \mathrm{Cov}_{u,w}[\tilde{f}, f^T 1 - \tilde{f}^T w], \quad \nabla_{u_m} \mathrm{D}_{\mathrm{KL}} = -w_m \mathrm{Cov}_{u,w} \left[ h(u_m), f^T 1 - \tilde{f}^T w \right], \tag{5}$$

where $h(\cdot, \theta) := \nabla_u f(\cdot, \theta)$, and the $\theta$ argument is again suppressed for brevity.

## 3.2 Stochastic optimization

The gradients in Eq. (5) involve expectations of (gradient) log-likelihoods from the model. Although there are a few particular Bayesian models where these can be evaluated in closed-form (e.g. the synthetic experiment in Section 4; see also Supp. C.1), this is not usually the case. In order to make the proposed pseudocoreset method broadly applicable, in this section we develop a black-box stochastic optimization scheme (Alg. 1) for Eq. (3).

To initialize the pseudocoreset, we subsample $M$ datapoints from the large dataset and reweight them to match the overall weight of the full dataset,

$$u_m \leftarrow x_{b_m}, \quad w_m \leftarrow N/M, \quad m = 1, \ldots, M$$
$$\mathcal{B} \sim \mathsf{UnifSubset} \left( [N], M \right), \quad \mathcal{B} := \{b_1, \ldots, b_M\}.$$

After initializing the pseudodata locations and weights, we simultaneously optimize Eq. (3) over both. Each optimization iteration $t \in \{1, \ldots, T\}$ consists of a stochastic gradient descent step with a learning rate $\gamma_t \propto t^{-1}$,

$$w_m \leftarrow \max \left( 0, w_m - \gamma_t (\hat{\nabla}_w)_m \right), \quad u_m \leftarrow u_m - \gamma_t \hat{\nabla}_{u_m}, \quad 1 \le m \le M.$$

---

**Algorithm 1** Pseudocoreset Variational Inference

---

1: **procedure** $\mathrm{PSVI}(f(\cdot,\cdot),\pi_0,x,M,B,S,T,(\gamma_t)_{t=1}^{\infty})$
    ▷ Initialize the pseudocoreset using a uniformly chosen subset of the full dataset
2:     $N \leftarrow$ # datapoints in $x$,    $\mathcal{B} \sim \mathsf{UnifSubset}\left([N],M\right)$,    $\mathcal{B} := \{b_1,\ldots,b_M\}$
3:     $u_m \leftarrow x_{b_m}$,    $w_m \leftarrow {}^{N}\!/\!{}_{M}$,    $m = 1,\ldots,M$
4:     **for** $t = 1,\ldots,T$ **do**
        ▷ Take $S$ samples from current pseudocoreset posterior
5:         $(\theta)_{s=1}^{S} \overset{\text{i.i.d.}}{\sim} \pi_{u,w}$ where $\pi_{u,w}(\theta) \propto \exp\left(\sum_{m=1}^{M} w_m f(u_m,\theta)\right)\pi_0(\theta)$
6:         $\mathcal{B} \sim \mathsf{UnifSubset}\left([N],B\right)$     ▷ Obtain a minibatch of $B$ datapoints from the full dataset
7:         **for** $s = 1,\ldots,S$ **do**           ▷ Compute (gradient) log-likelihood discretizations
8:             $g_s \leftarrow \left(f(x_b,\theta_s) - {}^{1}\!/\!{}_{S}\sum_{s'=1}^{S} f(x_b,\theta_{s'})\right)_{b\in\mathcal{B}} \in \mathbb{R}^B$
9:             $\tilde{g}_s \leftarrow \left(f(u_m,\theta_s) - {}^{1}\!/\!{}_{S}\sum_{s'=1}^{S} f(u_m,\theta_{s'})\right)_{m=1}^{M} \in \mathbb{R}^M$
10:             **for** $m = 1,\ldots,M$ **do**
11:                 $\tilde{h}_{m,s} \leftarrow \nabla_u f(u_m,\theta_s) - {}^{1}\!/\!{}_{S}\sum_{s'=1}^{S}\nabla_u f(u_m,\theta_{s'}) \in \mathbb{R}^d$
12:         $\hat{\nabla}_w \leftarrow -{}^{1}\!/\!{}_{S}\sum_{s=1}^{S}\tilde{g}_s\left({}^{N}\!/\!{}_{B}g_s^T 1 - \tilde{g}_s^T w\right)$    ▷ Compute Monte-Carlo gradients for $w$
13:         **for** $m = 1,\ldots,M$ **do**           and $(u_m)_{m=1}^{M}$
14:             $\hat{\nabla}_{u_m} \leftarrow -w_m {}^{1}\!/\!{}_{S}\sum_{s=1}^{S}\tilde{h}_{m,s}\left({}^{N}\!/\!{}_{B}g_s^T 1 - \tilde{g}_s^T w\right)$
15:         $w \leftarrow \max(w - \gamma_t\hat{\nabla}_w, 0)$           ▷ Take stochastic gradient step in $w$
16:         **for** $m = 1,\ldots,M$ **do**           and $(u_m)_{m=1}^{M}$
17:             $u_m \leftarrow u_m - \gamma_t\hat{\nabla}_{u_m}$
18:     **return** $w, (u_m)_{m=1}^{M}$

---

The stochastic gradient estimates $\hat{\nabla}_w \in \mathbb{R}^M$ and $\hat{\nabla}_{u_m} \in \mathbb{R}^d$ are based on $S \in \mathbb{N}$ samples $\theta_s \overset{\text{i.i.d.}}{\sim} \pi_{u,w}$ from the coreset approximation and a minibatch of $B \in \mathbb{N}$ datapoints from the full dataset,

$$\hat{\nabla}_w := -\frac{1}{S}\sum_{s=1}^{S}\tilde{g}_s\left(\frac{N}{B}g_s^T 1 - \tilde{g}_s^T w\right), \quad \hat{\nabla}_{u_m} := -w_m\frac{1}{S}\sum_{s=1}^{S}\tilde{h}_{m,s}\left(\frac{N}{B}g_s^T 1 - \tilde{g}_s^T w\right),$$

where

$$\tilde{h}_{m,s} := \nabla_u f(u_m,\theta_s) - \frac{1}{S}\sum_{s'=1}^{S}\nabla_u f(u_m,\theta_{s'}), \qquad g_s := \left.\left(f(\theta_s) - \frac{1}{S}\sum_{s'=1}^{S} f(\theta_{s'})\right)\right|_{\mathcal{B}}$$

$$\tilde{g}_s := \tilde{f}(\theta_s) - \frac{1}{S}\sum_{s'=1}^{S}\tilde{f}(\theta_{s'}), \qquad\qquad\qquad \mathcal{B} \sim \mathsf{UnifSubset}\left([N],B\right),$$

and $(\cdot)|_{\mathcal{B}}$ denotes restriction of a vector to only those indices in $\mathcal{B} \subset [N]$. Crucially, note that this computation does not scale with $N$, but rather with the number of coreset points $M$, the sample and minibatch sizes $S$ and $B$, and the dimension $d$. Obtaining $\theta_s \overset{\text{i.i.d.}}{\sim} \pi_{u,w}$ efficiently via Markov chain Monte Carlo sampling algorithms [21, 26] is (roughly) $O(M)$ per sample, because the coreset is always of size $M$; and we need not compute the entire vector $g_s \in \mathbb{R}^N$ per sample $s$, but rather only those $B \ll N$ indices in the minibatch $\mathcal{B}$, resulting in a cost of $O(B)$. Aside from that, all computations involving $\tilde{g}_s \in \mathbb{R}^M$ and $\tilde{h}_{m,s} \in \mathbb{R}^d$ are at most $O(Md)$. Each of these computations are repeated $S$ times over the coreset posterior samples.

### 3.3 Differentially Private Scheme

Beyond better summarizations of high-dimensional data, pseudocoresets enable the generation of a data summarization that ensures the statistical privacy of individual datapoints under the model of (approximate) *differential privacy*. In this setting, a trusted curator holds an aggregate dataset of $N$ datapoints, $x \in \mathcal{X}^N$, $\mathcal{X} \subseteq \mathbb{R}^d$, and builds and releases a pseudocoreset $(u,w)$, $u \in \mathcal{X}^M$, $w \in \mathbb{R}_+^M$ via a randomized mechanism satisfying Definition 2 [14, 13].

**Definition 2** (($\varepsilon,\delta$)-Differentially Private Coreset)**.** Fix $\varepsilon \geq 0, \delta \in [0,1]$. A pseudocoreset construction algorithm $\mathcal{M}: \mathcal{X}^N \to \mathbb{R}_+^M \times \mathcal{X}^M$ is ($\varepsilon,\delta$)-differentially private if for every pair of adjacent datasets $x \approx x'$ and all events $A \subseteq \mathbb{R}_+^M \times \mathcal{X}^M$, $\mathbb{P}[\mathcal{M}(x) \in A] \leq e^{\varepsilon}\mathbb{P}[\mathcal{M}(x') \in A] + \delta$.

We consider two datasets $x, x'$ as adjacent (denoted $x \approx x'$) if $x'$ can be obtained from $x$ by adding or removing an element. $\varepsilon$ controls the effect that removal or addition of an element can have on the output distribution of $\mathcal{M}$, while $\delta$ captures the failure probability, and is preferably $o(1/N)$.

In this section, we develop a differentially private version of pseudocoreset construction. Beyond modifying our initialization scheme, private pseudocoreset construction comes as natural extension of Alg. 1 via replacing gradient computation involving points of the true dataset with its differentially private counterpart.

**Pseudodata points initialization**    In the standard (nonprivate) pseudocoreset construction (Alg. 1), pseudopoints are initialized from the dataset itself, incurring a privacy penalty. In differentially private pseudocoreset construction, we simply initialize pseudopoints by generating synthetic data from the statistical model at no privacy cost.

**Optimization**    Examining lines 4–19 of Alg. 1, the only steps that involve handling the original data occur at lines 8, 12, and 14, when we use the minibatch subsample to compute log-likelihoods and gradients. Due to the post-processing property of differential privacy [15], all of the other computations in Alg. 1 (e.g. sampling from the pseudocoreset posterior, computing pseudopoint log-likelihoods, etc.) incur no privacy cost. Therefore, we need only to control the influence of private data entering the gradient computation through the vector of $(g_s^T 1)_{s=1}^S$ terms.

To accomplish this we do repeated applications of the *subsampled Gaussian mechanism*, since this also allows us to use a *moments accountant* technique to keep tight estimates of privacy parameters [1, 41]. As in the nonprivate scheme, in each optimization step we uniformly subsample a minibatch $\mathcal{B} = \{x_1, \ldots, x_B\}$ of private datapoints. We then replace the $g_s^T 1$ term in lines 12 and 14 with a randomized privatization:

$$\text{replace} \quad (g_s^T 1)_{s=1}^S \quad \text{with} \quad Z + \sum_{i=1}^B \frac{G_i}{\max\left(1, \frac{||G_i||_2}{C}\right)}, \qquad Z \sim \mathcal{N}(0, \sigma^2 C^2 I), \qquad (6)$$

where $G_i := \left( f(x_i, \theta_s) - \frac{1}{S} \sum_{s'=1}^S f(x_i, \theta_{s'}) \right)_{s=1}^S \in \mathbb{R}^S \ \forall x_i \in \mathcal{B}$, and $C, \sigma > 0$ are parameters controlling the amount of privacy. This modification to Alg. 1 has been shown in past work to obtain the privacy guarantee provided in Corollary 3; crucially, the privacy cost of our construction is independent of the pseudocoreset size. It also does not introduce any significant amount of additional computation. No sensitivity computation for privatisation noise calibration is required, as boundedness is enforced via clipping in Eq. (6). Finally, a manageable number of privacy specific hyperparameters is introduced: the clipping bound $C$ and noise level $\sigma$.

**Corollary 3** ([1]). *There exist constants $c_1, c_2$ such that Alg. 1 modified per Eq. (6) is $(\varepsilon, \delta)$-differentially private for any $\varepsilon < c_1 q^2 T$, $\delta > 0$, and $\sigma \geq c_2 q \sqrt{T \log(1/\delta)}/\varepsilon$, where $q := \frac{B}{N}$ is the fraction of data in a minibatch and $T$ is the number of optimization steps.*

## 4   Experimental Results

In this section, we evaluate the posterior approximation quality achieved by pseudocoreset sparse VI (PSVI) compared against uniform random subsampling (Uniform), Hilbert coresets (GIGA [8]) and SparseVI greedy coreset construction [7]. For black-box constructions of SparseVI and PSVI we used $S = 100$ Monte Carlo samples per gradient estimation. For GIGA we used a 100-dimensional random projection from a Gaussian approximate posterior $\hat{\pi}$ with two choices for mean and covariance: one set to the exact posterior (Optimal), which is not tractable to obtain in practice and forms an optimistic estimate of achievable approximation quality; and one with mean and covariance set to a random point on the interpolant between the prior and the exact posterior point estimates, and subsequently corrupted with 75% additive relative noise (Realistic). Notably, Hilbert coresets and SparseVI develop incremental schemes for construction, while PSVI relies on batch optimization with random initialization (Alg. 1), and does not use any information from pseudocoresets of smaller size. An incremental scheme for SparseVI is included in Supp. C. Code for the presented experiments is available at https://github.com/trevorcampbell/pseudocoresets-experiments.

**Gaussian mean inference**    We first evaluate the performance of PSVI on a synthetic dataset of $N = 10^3$ datapoints, where we aim to infer the posterior mean $\theta \sim \mathcal{N}(\mu_0, \Sigma_0)$ of a $d$-dimensional Gaussian conditioned on Gaussian observations $(X_n)_{n=1}^N \overset{\text{i.i.d.}}{\sim} \mathcal{N}(\theta, \Sigma)$. In this example, the exact

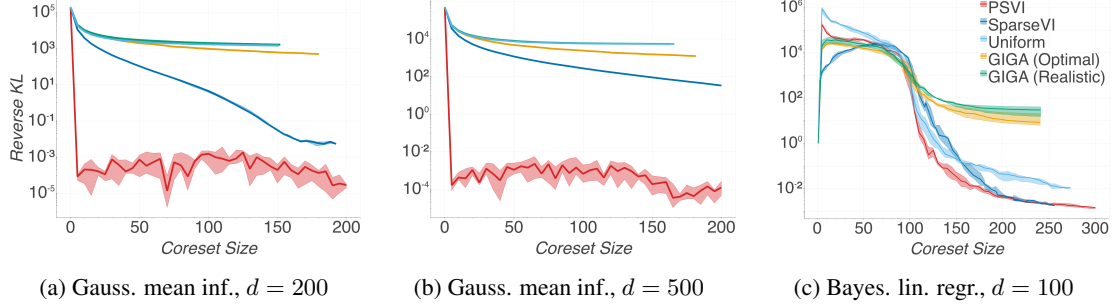

|          |          |          |
|:--------:|:--------:|:--------:|
| (a) Gauss. mean inf., $d = 200$ | (b) Gauss. mean inf., $d = 500$ | (c) Bayes. lin. regr., $d = 100$ |

Figure 2: Comparison of (pseudo)coreset approximate posterior quality for experiments on synthetic datasets over 10 trials. Solid lines display the median KL divergence, with shaded areas showing 25th and 75th percentiles of KL divergence. In Fig. 2c, KL divergence is normalized by the prior.

pseudocoreset posterior for any set of weights $(w_m)_{m=1}^M$ and pseudopoint locations $(u_m)_{m=1}^M$ is available in closed-form:

$$\Sigma_{u,w} = (\Sigma_0^{-1} + \sum_{m=1}^M w_m \Sigma^{-1})^{-1} \qquad \mu_{u,w} = \Sigma_{u,w}(\Sigma_0^{-1}\mu_0 + \Sigma^{-1}\sum_{m=1}^M w_m u_m).$$

Using the exact posterior, we derive the exact moments used in the gradient formulae from Eq. (5) in closed form (see Supp. C.1),

$$\text{Cov}_{u,w}[f_n, f_m] = v_n^T \Psi v_m + 1/2 \operatorname{tr} \Psi^T \Psi, \quad \text{Cov}_{u,w}[\tilde{f}_n, f_m] = \tilde{v}_n^T \Psi v_m + 1/2 \operatorname{tr} \Psi^T \Psi,$$

$$\text{Cov}_{u,w}[h(u_i), f_n] = Q^{-T}\Psi v_n, \qquad\qquad \text{Cov}_{u,w}[h(u_i), \tilde{f}_n] = Q^{-T}\Psi \tilde{v}_n,$$

where $Q$ is the lower triangular matrix of the Cholesky decomposition of $\Sigma$ (i.e. $\Sigma = QQ^T$), $\Psi := Q^{-1}\Sigma_{u,w}Q^{-T}$, $v_n := Q^{-1}(x_n - \mu_{u,w})$, and $\tilde{v}_m := Q^{-1}(u_m - \mu_{u,w})$. We vary the pseudo-coreset size from $M = 1$ to 200, and set the total number of iterations to $T = 500$. We use learning rates $\gamma_t(M) = \alpha(M)t^{-1}$, where $\alpha(M) = 1$ for SparseVI and $\alpha(M) = \max(1.1 - 0.005M, 0.2)$ for PSVI. As verified in Figs. 2a and 2b, Hilbert coresets provide poor quality summarizations in the high-dimensional regime, even for large coreset sizes. Despite showing faster decrease of approximation error for a larger range of coreset sizes, SparseVI is also fundamentally limited by the use of the original datapoints, per Proposition 1. Furthermore, we observe that the quality of all previous coreset methods when $d = 500$ is significantly lower compared to $d = 200$. On the other hand, the KL divergence for PSVI decreases significantly more quickly, giving a near perfect approximation for the true posterior with a single pseudodata point, regardless of data dimension. As shown earlier in Fig. 1a, PSVI has the capacity to move the pseudodata points in order to capture the true posterior very efficiently.

**Bayesian linear regression** In the second experiment, we use a set of $N = 2,000$ 101-dimensional datapoints $(x_n, y_n)_{n=1}^N$ generated as follows:

$$(x_n)_{n=1}^N \overset{\text{i.i.d.}}{\sim} \mathcal{N}(0, I), \quad (y_n)_{n=1}^N \sim [1, x_n]^T\theta + \epsilon_n, \quad (\epsilon_n)_{n=1}^N \overset{\text{i.i.d.}}{\sim} \mathcal{N}(0, \sigma^2),$$

and aim to infer $\theta \in \mathbb{R}^{101}$. We assume a prior $\theta \sim \mathcal{N}(\mu_0, \sigma_0^2 I)$, where $\mu_0, \sigma_0^2$ are the dataset empirical mean and second moment, and set the noise parameter $\sigma$ to the variance of $(y_n)_{n=1}^N$. We apply stochastic optimization for PSVI construction (also see Supp. C.2.1). We use learning rates $\gamma_t = t^{-1}$ for SparseVI, and $\gamma_t = 0.1t^{-1}$ for PSVI, $B = 200$, $T = 1000$, while selection step for SparseVI is carried out over the full dataset. Fig. 2c shows that Hilbert coresets cannot improve posterior approximation in this setting with 100 random projections (see Supp. C.2.2), while PSVI achieves the fastest decay rate over sizes $100 \leq M < 250$, surpassing SparseVI.

**Bayesian logistic regression** Finally, we compare (pseudo)coreset construction methods on Bayesian logistic regression applied to 3 large (8.4–100$K$ datapoints, 50–237 dimensions) datasets. For brevity, equations and gradients for the logistic regression model, additional experiments on 3 smaller-scale datasets, full dataset descriptions, hyperparameter selection, time performance evaluation and results on an incremental scheme for pseudocoreset construction are deferred to Supp. C.3. For PSVI and SparseVI we use minibatch size $B = 200$, number of gradient updates $T = 500$, and learning rate schedules $\gamma_t = \alpha t^{-1}$. For TRANSACTIONS, CHEMREACT100 and MUSIC, $\alpha$ is respectively set to 0.1, 0.1, 10 for SparseVI, and 1, 10, 10 for PSVI. In the selection step of

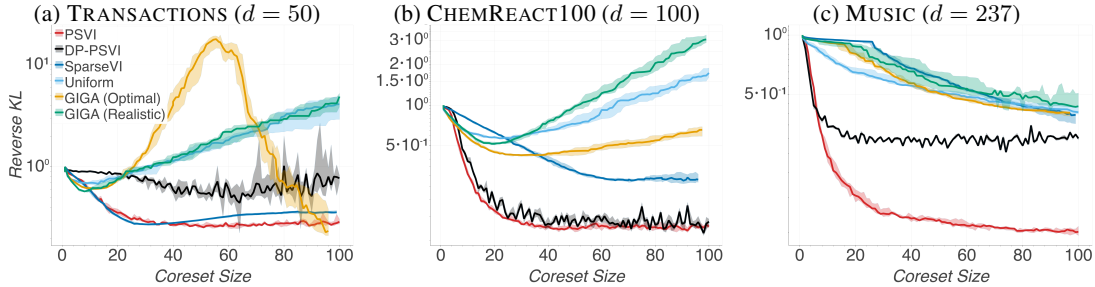

Figure 3: Comparison of (pseudo)coreset approximate posterior quality vs coreset size for logistic regression over 10 trials on 3 large-scale datasets. Presented differentially private pseudocoresets correspond to $(0.2, 1/N)$-DP. Reverse KL divergence is displayed normalized by the prior.

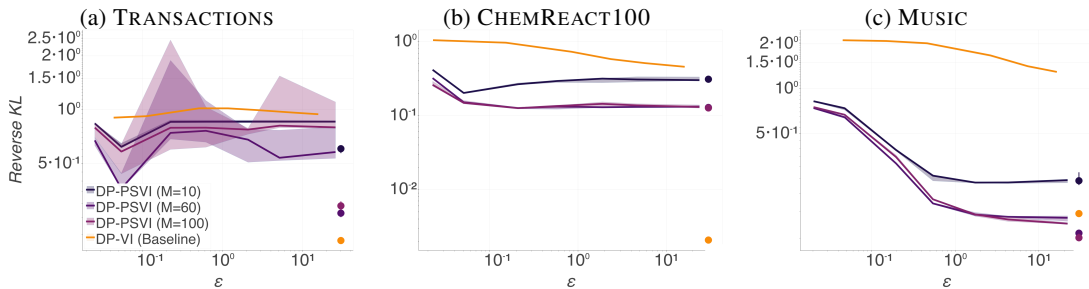

Figure 4: Approximate posterior quality over decreasing differential privacy guarantees for private pseudocoresets of varying size (DP-PSVI) plotted against private variational inference (DP-VI, [27]). $\delta$ is always kept fixed at $1/N$. Markers on the right end of each plot display the errorbar of approximation achieved by the corresponding nonprivate posteriors. Results are displayed over 5 trials for each construction.

SparseVI we use a uniform subsample of $1,000$ datapoints. For the differentially private pseudocoreset constructions (DP-PSVI), we use a subsampling ratio $q = 2 \times 10^{-3}$. At each iteration we adapt the clipping norm value $C$ to the median norm of $(f(u_m, \theta_s) - \frac{1}{S}\sum_{s'=1}^{S} f(u_m, \theta_{s'}))_{s=1}^{S}$ computed over pseudodata point values $u_m$, and use noise level $\sigma = 5$. Our hyperparameters choice implies privacy parameters $\varepsilon = 0.2$ and $\delta = 1/N$ for each of the datasets. We initialise each pseudocoreset of size $M$ via sampling $(x_m)_{m=1}^{M} \overset{\text{i.i.d.}}{\sim} \mathcal{N}(0, I)$, and sampling $\theta$, $(y_m)_{m=1}^{M}$ from the statistical model. Results presented in Fig. 3 demonstrate that PSVI achieves consistently the smallest posterior approximation error in the small coreset size regime, offering improvement compared to SparseVI and being competitive with GIGA (Optimal), without the requirement for specifying a weighting function. In Fig. 3a, for $M \geq d$ GIGA (Optimal) follows a much steeper decrease in KL divergence, reflecting the dependence of its approximation quality on dataset dimension per Proposition 1. In contrast, PSVI typically reaches its minimum at $M < d$. The difference in approximation quality becomes clearer in higher dimensions (e.g. MUSIC, where $d = 237$). Perhaps surprisingly, the private pseudocoreset construction has only marginally worse approximation quality compared to nonprivate PSVI and generally achieves better peformance in comparison to the other state-of-the-art nonprivate coreset constructions.

In Fig. 4 we present the achieved posterior approximation quality via DP-PSVI, against a competitive state-of-the-art method for general-purpose private inference (DP-VI, [27]). The plots display the behaviour of methods over a wide range of $\varepsilon$ values, achieved using varying levels of privatization noise, and $\delta$ always set to $1/N$. For logistic regression, DP-VI infers an approximate posterior from the family of Gaussians with diagonal covariance via ADVI [28], followed by an additional Laplace approximation. Note that by design, DP-VI is constrained by the usual Gaussian variational approximation, while DP-PSVI is more flexible and can approach the true posterior as $M$ increases—this effect is reflected in nonprivate posteriors as well, as data dimensionality grows (see for example Fig. 4c). Indeed, we verify that in the high-privacy regime DP-PSVI, for sufficient pseudocoreset size (which is typically small for tested real-world datasets), offers posterior approximation with better KL divergence compared to DP-VI. Our findings indicate that private PSVI offers

efficient releases of big data via informative pseudopoints, which enable arbitrary post processing (e.g. running any *nonprivate* black-box algorithm for Bayesian inference), under strong privacy guarantees and without reducing the quality of inference.

## 5 Conclusion

We introduced a new variational formulation for Bayesian coreset construction, which yields efficient summarizations for big and high-dimensional datasets via simultaneously learning pseudodata points locations and weights. We proved limitations of existing variational formulations for coresets and demonstrated that they can be resolved with our new methodology. We proposed an efficient construction scheme via black-box stochastic optimization and showed how it can be adapted for differentially private Bayesian summarization. Finally, we demonstrated the applicability of our methodology on synthetic and real-world datasets, and practical statistical models.

## Broader Impact

Pseudocoreset variational inference is a general-purpose Bayesian inference algorithm, hence shares implications mostly encountered in approximate inference methods. For example, replacing the full dataset with a pseudocoreset has the potential to cause inferential errors; these can be partially tempered by using a pseudocoreset of larger size. Note also that the optimization algorithm in this work aims to reduce KL divergence: however the proposed variational objective might be misleading in many applications and lead to incorrect conclusions in certain statistical models (e.g. point estimates and uncertainties might be far off despite KL being almost zero [24]). Moreover, Bayesian inference in general is prone to model misspecification. Therefore, a pseudocoreset summarization based on a wrong statistical model will lead to non-representative compression for inferential purposes. Constructing the coreset on a statistical model suited for robust inference instead of the original one [31, 40], can offer protection against modelling mismatches. Naturally, the utility of generated dataset summary becomes task-dependent, as it has been optimized for a specific learning objective, and cannot be fully transferable to multiple different inference tasks on the same dataset.

Our learnable pseudodata are also generally not as interpretable as the points of previous coreset methods, as they are not real data. And the level of interpretability is model specific. This creates a risk of misinterpretation of pseudocoreset points in practice. On the other hand, our optimization framework does allow the introduction of interpretability constraints (e.g. pseudodata sparsity) to explicitly capture interpretability requirements.

Pseudocoreset-based summarization is susceptible to reproducing potential biases and unfairness existing in the original dataset. Majority-group datapoints in the full dataset which capture information relevant to the statistical task of interest are expected to remain over-represented in the learned summary; while minority-group datapoints might be eliminated, if their distinguishing features are not related to inference. Amending the initialization step to contain such datapoints, or using a prior that strongly favors a debiased version of the dataset, could both mitigate these concerns; but more study is warranted.

## Acknowledgments and Disclosure of Funding

ZX and TC were supported by a National Sciences and Engineering Research Council of Canada (NSERC) Discovery Grant and an NSERC Discovery Launch Supplement. DM and CM were partially supported by Nokia Bell Labs through their donation for the Centre of Mobile, Wearable Systems and Augmented Intelligence to the University of Cambridge. DM also gratefully acknowledges the financial support from Lundgren Fund and Darwin College Cambridge. The authors thank Rik Sarkar for useful conversations in the early stages of this work.

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
