[Supplementary Material]

# A Technical Results and Proofs

In the setting of Proposition 1, both the exact posterior and the coreset posterior are multivariate Gaussian distributions, denoted as $\mathcal{N}(\mu_1, \Sigma_1)$ and $\mathcal{N}(\mu_w, \Sigma_w)$ respectively. The mean and covariance are

$$\Sigma_1 = \frac{1}{1+N} I_d, \quad \mu_1 = \Sigma_1 \left( \sum_{n=1}^{N} X_n \right), \tag{7}$$

and

$$\Sigma_w = \frac{I_d}{1 + \left( \sum_{n=1}^{N} w_n \right)}, \quad \mu_w = \Sigma_w \left( \sum_{n=1}^{N} w_n X_n \right). \tag{8}$$

*Proof of Proposition 1.* By Eqs. (7) and (8),

$$D_{KL}(\pi_w \| \pi) = \frac{1}{2} \left[ \log \frac{|\Sigma_1|}{|\Sigma_w|} - d + \mathrm{tr}\left(\Sigma_1^{-1} \Sigma_w\right) + (\mu_1 - \mu_w)^T \Sigma_1^{-1}(\mu_1 - \mu_w) \right]$$

$$= \frac{1}{2} \left[ -d \log \left( \frac{1+N}{1 + \sum_{n=1}^{N} w_n} \right) - d + d \left( \frac{1+N}{1 + \sum_{n=1}^{N} w_n} \right) + (\mu_1 - \mu_w)^T \Sigma_1^{-1}(\mu_1 - \mu_w) \right].$$

Note that $\forall x > 0, x - 1 \geq \log x$, implying that

$$-d \log \left( \frac{1+N}{1 + \sum_{n=1}^{N} w_n} \right) - d + d \left( \frac{1+N}{1 + \sum_{n=1}^{N} w_n} \right) \geq 0.$$

Thus,

$$D_{KL}(\pi_w \| \pi) \geq \frac{1}{2}(\mu_1 - \mu_w)^T \Sigma_1^{-1}(\mu_1 - \mu_w).$$

Suppose we pick a set $\mathcal{I} \subseteq [N], |\mathcal{I}| = M$ of active indices $n$ where the optimal $w_n \geq 0$, and enforce that all others $n \notin \mathcal{I}$ satisfy $w_n = 0$. Then denoting

$$Y = [X_n : n \notin \mathcal{I}] \in \mathbb{R}^{d \times (N-M)}, \quad X = [X_n : n \in \mathcal{I}] \in \mathbb{R}^{d \times M},$$

we have that for any $w \in \mathbb{R}_+^M$ for those indices $\mathcal{I}$,

$$D_{KL}(\pi_w \| \pi) \geq \frac{1}{2(N+1)} 1^T Y^T Y 1 + 1^T Y^T X \left( \frac{1}{N+1} - \frac{w}{1 + 1^T w} \right)$$

$$+ \frac{N+1}{2} \left( \frac{1}{N+1} - \frac{w}{1 + 1^T w} \right)^T X^T X \left( \frac{1}{N+1} - \frac{w}{1 + 1^T w} \right).$$

Relaxing the nonnegativity constraint, replacing $w/(1 + 1^T w)$ with a generic $x \in \mathbb{R}^M$, and noting that $X^T X$ is invertible almost surely when $M < d$, we can optimize this expression yielding a lower bound on the optimal KL divergence using active index set $\mathcal{I}$,

$$D_{KL}\left(\pi_{w_\mathcal{I}^\star} \| \pi\right) \geq \frac{1^T Y^T \left(I - X(X^T X)^{-1} X^T\right) Y 1}{2(N+1)}.$$

The numerator is the squared norm of $Y1$ minus its projection onto the subspace spanned by the $M$ columns of $X$. Since $Y1 \sim \mathcal{N}(0, (N-M)I), Y1 \in \mathbb{R}^d$ is an isotropic Gaussian, then its projection onto the orthogonal complement of any fixed subspace of dimension $M$ is also an isotropic Gaussian of dimension $d - M$ with the same variance. Since the columns of $X$ are also independent and isotropic, its column subspace is uniformly distributed. So therefore, for each possible choice of $\mathcal{I}$

$$D_{KL}\left(\pi_{w_\mathcal{I}^\star} \| \pi\right) \geq \frac{N-M}{2(N+1)} Z_\mathcal{I}, \quad Z_\mathcal{I} \sim \chi^2(d-M).$$

Note that the $Z_\mathcal{I}$ will have dependence across the $\binom{N}{M}$ different choices of index subset $\mathcal{I}$. Thus, the probability that *all* $Z_\mathcal{I}$ are large is

$$\mathbb{P}\left( \min_{\mathcal{I} \subseteq [N], |\mathcal{I}|=M} Z_\mathcal{I} > \epsilon \right) \geq 1 - \binom{N}{M} \mathbb{P}(Z_\mathcal{I} \leq \epsilon)$$

$$= 1 - \binom{N}{M} F_{d-M}(\epsilon),$$

where $F_k$ is the CDF for the $\chi^2$ distribution with $k$ degrees of freedom. The result follows. $\square$

## B  Gradient Derivations

Throughout, expectations and covariances over the random parameter $\theta$ with no explicit subscripts are taken under pseudocoreset posterior $\pi_{u,w}$. We also interchange differentiation and integration without explicitly verifying that sufficient conditions to do so hold.

### B.1  Weights gradient

First, we compute the gradient with respect to weights vector $w \in \mathbb{R}_+^M$, which is written as

$$\nabla_w \mathrm{D_{KL}} = -\nabla_w \log Z(u,w) - \nabla_w \mathbb{E}[f(\theta)^T 1] + \nabla_w \mathbb{E}[\tilde{f}(\theta)^T w].$$

For any function $a : \Theta \to \mathbb{R}$, we have that

$$\nabla_w \mathbb{E}\left[a(\theta)\right] = \int \nabla_w \left( \exp\left( w^T \tilde{f}(\theta) - \log Z(u,w) \right) \right) a(\theta) \pi_0(\theta) \mathrm{d}\theta$$
$$= \mathbb{E}\left[ \left( \tilde{f}(\theta) - \nabla_w \log Z(u,w) \right) a(\theta) \right].$$

Next, we compute the gradient of the log normalization constant via

$$\nabla_w \log Z(u,w) = \int \frac{1}{Z(u,w)} \nabla_w \left( \exp\left( w^T \tilde{f}(\theta) \right) \right) \pi_0(\theta) \mathrm{d}\theta$$
$$= \mathbb{E}\left[ \tilde{f}(\theta) \right].$$

Combining, we have

$$\nabla_w \mathbb{E}\left[a(\theta)\right] = \mathbb{E}\left[ \left( \tilde{f}(\theta) - \mathbb{E}\left[ \tilde{f}(\theta) \right] \right) a(\theta) \right].$$

Subtracting $0 = \mathbb{E}\left[a(\theta)\right] \mathbb{E}\left[ \tilde{f}(\theta) - \mathbb{E}\left[ \tilde{f}(\theta) \right] \right]$ yields

$$\nabla_w \mathbb{E}\left[a(\theta)\right] = \mathrm{Cov}\left[ \tilde{f}(\theta), a(\theta) \right].$$

The gradient with respect to $w$ in Eq. (5) follows by substituting $1^T f(\theta)$ and $w^T \tilde{f}(\theta)$ for $a(\theta)$ and using the product rule.

### B.2  Location gradients

Here we take the gradient with respect to a single pseudopoint $u_i \in \mathbb{R}^d$. First note that

$$\nabla_{u_i} \mathrm{D_{KL}} = -\nabla_{u_i} \log Z(u,w) - \nabla_{u_i} \mathbb{E}[f(\theta)^T 1] + \nabla_{u_i} \mathbb{E}[\tilde{f}(\theta)^T w].$$

For any function $a(u,\theta) : \mathbb{R}^{d \times M} \times \Theta \to \mathbb{R}$, we have

$$\nabla_{u_i} \mathbb{E}\left[a(u,\theta)\right] = \int \nabla_{u_i} \left( \exp\left( w^T \tilde{f}(\theta) - \log Z(u,w) \right) a(u,\theta) \right) \pi_0(\theta) \mathrm{d}\theta.$$

Using the product rule and recalling from the main text that $h(\cdot,\theta) := \nabla_u f(\cdot,\theta)$,

$$\nabla_{u_i} \mathbb{E}\left[a(u,\theta)\right] = \mathbb{E}\left[\nabla_{u_i} a(u,\theta)\right] + \mathbb{E}\left[a(u,\theta)\left(w_i h(u_i,\theta) - \nabla_{u_i} \log Z(u,w)\right)\right].$$

Taking the gradient of the log normalization constant using similar techniques,

$$\nabla_{u_i} \log Z(u,w) = w_i \mathbb{E}\left[h(u_i,\theta)\right].$$

Combining,

$$\nabla_{u_i} \mathbb{E}\left[a(u,\theta)\right] = \mathbb{E}\left[\nabla_{u_i} a(u,\theta)\right] + w_i \mathbb{E}\left[a(u,\theta)\left(h(u_i,\theta) - \mathbb{E}\left[h(u_i,\theta)\right]\right)\right].$$

Subtracting $0 = \mathbb{E}\left[a(u,\theta)\right]\mathbb{E}\left[(h(u_i,\theta) - \mathbb{E}\left[h(u_i,\theta)\right])\right]$ yields

$$\nabla_{u_i}\mathbb{E}\left[a(u,\theta)\right] = \mathbb{E}\left[\nabla_{u_i}a(u,\theta)\right] + w_i\,\mathrm{Cov}\left[a(u,\theta), h(u_i,\theta)\right].$$

The gradient with respect to $u_i$ in Eq. (5) follows by substituting $f(\theta)^T 1$ and $\tilde{f}(\theta)^T w$ for $a(u,\theta)$.

## C    Details on Experiments

### C.1    Gaussian mean inference

Let the coreset posterior have mean $\mu_{u,w}$ and covariance matrix $\Sigma_{u,w}$. Throughout, expectations and covariances over the random parameter $\theta$ with no explicit subscripts are taken under pseudocoreset posterior $\pi_{u,w}$. Define $\Psi := Q^{-1}\Sigma_{u,w}Q^{-T}$, $v_n := Q^{-1}(x_n - \mu_{u,w})$, $\tilde{v}_n := Q^{-1}(u_n - \mu_{u,w})$, and $Q$ to be the lower triangular matrix of the Cholesky decomposition of $\Sigma$, i.e. $\Sigma := QQ^T$. In order to compute the gradients in Eq. (5), we need expressions for $\mathrm{Cov}[f_n, f_m]$, $\mathrm{Cov}[\tilde{f}_n, f_m]$, $\mathrm{Cov}[h(u_i), f_n]$, and $\mathrm{Cov}[h(u_i), \tilde{f}_n]$.

Following [1], we have that

$$\mathrm{Cov}[f_n, f_m] = v_n^T \Psi v_m + \frac{1}{2}\,\mathrm{tr}\,\Psi^T\Psi$$

$$\mathrm{Cov}[\tilde{f}_n, f_m] = \tilde{v}_n^T \Psi v_m + \frac{1}{2}\,\mathrm{tr}\,\Psi^T\Psi.$$

We now evaluate the remaining covariance $\mathrm{Cov}[h(u_i), f_m]$; the derivation of $\mathrm{Cov}[h(u_i), \tilde{f}_m]$ follows similarly. We begin by explicitly evaluating the log-likelihood gradient and its expectation,

$$h(u_i) = -\Sigma^{-1}(u_i - \theta)$$

$$\mathbb{E}\left[h(u_i)\right] = -\Sigma^{-1}(u_i - \mu_{u,w}),$$

and again following [1], we have (up to a constant) that

$$f_n = -\frac{1}{2}(x_n - \theta)^T\Sigma^{-1}(x_n - \theta)$$

$$\mathbb{E}\left[f_n\right] = -\frac{1}{2}\,\mathrm{tr}\,\Psi - \frac{1}{2}\|v_n\|^2.$$

Thus using the above definitions,

$$\mathbb{E}\left[h(u_i)\right]\mathbb{E}\left[f_n\right] = \frac{\left(\mathrm{tr}\,\Psi + \|v_n\|^2\right)}{2}Q^{-T}\tilde{v}_i.$$

Next,

$$\mathbb{E}\left[h(u_i)f_n\right] = \frac{1}{2}\Sigma^{-1}\mathbb{E}\left[(u_i - \theta)(x_n - \theta)^T\Sigma^{-1}(x_n - \theta)\right].$$

Defining $z \sim \mathcal{N}(0, \Psi)$, and using the above definitions,

$$\mathbb{E}\left[h(u_i)f_n\right] = \frac{1}{2}Q^{-T}\mathbb{E}\left[(\tilde{v}_i - z)(v_n - z)^T(v_n - z)\right].$$

Evaluating the expectation, noting that odd order moments of $z$ are equal to 0,

$$\mathbb{E}\left[h(u_i)f_n\right] = \frac{\|v_n\|^2 + \mathrm{tr}\,\Psi}{2}Q^{-T}\tilde{v}_i + Q^{-T}\Psi v_n.$$

Therefore,

$$\mathrm{Cov}[h(u_i), f_n] = Q^{-T}\Psi v_n,$$

|  | (a) project. dim. $= 200$ | (b) project. dim. $= 2,000$ | (c) project. dim. $= 10,000$ |

Figure 5: Comparison of Hilbert coresets performance on Bayesian linear regression experiment for increasing projection dimension (over 10 trials).

and likewise,

$$\text{Cov}[h(u_i), \tilde{f}_n] = Q^{-T} \Psi \tilde{v}_n.$$

## C.2 Bayesian linear regression

### C.2.1 Model and gradients details

Here we present the terms involving pseudodata points—the corresponding expressions for original datapoints are the same, after replacing $u_m$ with $x_m$.

For individual points, dropping normalization constants, we get log-likelihood terms of the form

$$f_m(\theta) = -\frac{1}{2\sigma^2} \left( y_m - \theta^T u_m \right)^2.$$

Hence, we obtain for the pseudocoreset posterior

$$\pi_{u,w} = \mathcal{N}(\mu_{u,w}, \Sigma_{u,w}), \quad \text{where}$$

$$\Sigma_{u,w} = (\sigma_0^{-2} I + \sigma^{-2} \sum_{m=1}^{M} w_m u_m u_m^T)^{-1}, \quad \mu_{u,w} = \Sigma_{u,w} (\sigma_0^{-2} I \mu_0 + \sigma^{-2} \sum_{m=1}^{M} w_m y_m u_m).$$

To scale up computation on large datasets, in our experiment we made use of stochastic gradients for black-box construction of `PSVI` and `SparseVI`. Beyond the expressions for individual log-likelihood and (pseudo)coreset posteriors presented above, for pseudocoreset construction we also need the expression for log-likelihood gradient with respect to the pseudodata points, for which we can immediately see that $\nabla_{u_m} f(u_m, \theta) = \frac{1}{\sigma^2}(y_m - \theta^T u_m)\theta$. Over our experiment, we optimized initial learning rates for `SparseVI` and `PSVI` via a grid search over $\{0.1, 1, 10\}$.

### C.2.2 Additional plots

Here we present some more plots demonstrating the dependence of Hilbert coresets approximation quality on the dimension of random projections in the Bayesian linear regression setting presented in Fig. 2c. We remind that the dimension used at this experiment and throughout the entire experiments section was set to 100. Increasing this number is typically expensive to obtain in practice. As demonstrated in Fig. 5, getting higher projection dimension enables better posterior approximation in the problem, for both `GIGA (Optimal)` and `GIGA (Realistic)`. However, `PSVI` remains competitive in the small coreset regime even for Hilbert coresets with extremely large projection dimensionality, demonstrating the information-geometric limitations that Hilbert coreset constructions are known to face [1].

### C.3 Bayesian Logistic Regression

#### C.3.1 Model

In logistic regression we have a set of datapoints $(x_n, y_n)_{n=1}^N$ each corresponding to a feature vector $x_n \in \mathbb{R}^d$ and a label $y_n \in \{-1, 1\}$. Datapoints are assumed to be generated according to the following statistical model

$$y_n | x_n, \theta \sim \text{Bern}\left(\frac{1}{1 + e^{-z_n^T \theta}}\right) \quad z_n := \begin{bmatrix} x_n \\ 1 \end{bmatrix}.$$

The aim of inference is to compute the posterior over the latent parameter $\theta = [\theta_0 \ldots \theta_d]^T \in \mathbb{R}^{d+1}$. The log-likelihood of each datapoint can be expressed as

$$
\begin{aligned}
f_n(x_n, y_n | \theta) =& \mathbb{1}[y_n = -1] \log\left(1 - \frac{1}{1 + e^{-z_n^T \theta}}\right) - \mathbb{1}[y_n = 1] \log\left(1 + e^{-z_n^T \theta}\right) \\
=& -\log\left(1 + \exp(-y_n z_n^T \theta)\right).
\end{aligned}
$$

Hence in pseudocoreset construction we can optimize pseudodata point locations with respect to continuous variable $x_n$, using the gradient

$$\nabla_{x_n} f_n = \frac{e^{-y_n z_n^T \theta}}{1 + e^{-y_n z_n^T \theta}} y_n \begin{bmatrix} \theta_1 \\ \vdots \\ \theta_d \end{bmatrix}.$$

#### C.3.2 Datasets description

For logistic regression experiments, we used subsampled and full versions of datasets presented in Table 1: a synthetic dataset with $x \in \mathbb{R}^2$ sampled i.i.d. from a $\mathcal{N}(0, I)$ and $y \in \{-1, 1\}$ sampled from respective logistic likelihood with $\theta = [3, 3, 0]^T$ (SYNTHETIC); a phishing websites dataset reduced to $D = 10$ via PCA (PHISHING); a chemical reactivity dataset with real-valued features corresponding to its first 10 and 100 principal components (CHEMREACT and CHEMREACT100 respectively); a dataset with 50 real-valued features associated with whether each of $100K$ customers of a bank will make a specific transaction (TRANSACTIONS); and a dataset for music analysis, where we consider the "classical vs all" genre classification task (MUSIC). Original versions of the four latter datasets are available online respectively at https://www.csie.ntu.edu.tw/~cjlin/libsvmtools/datasets/binary.html, http://komarix.org/ac/ds, https://www.kaggle.com/c/santander-customer-transaction-prediction/data, and https://github.com/mdeff/fma.

| Dataset name | $N$ | $D$ |
|---|---|---|
| SYNTHETIC | 500 | 2 |
| PHISHING | 500 | 10 |
| CHEMREACT | 500 | 10 |
| TRANSACTIONS | 100,000 | 50 |
| CHEMREACT100 | 26,733 | 100 |
| MUSIC | 8,419 | 237 |

Table 1: Details for datasets used in logistic regression experiments.

#### C.3.3 Small-scale experiments

In the small-scale experiment, the number of overall gradient updates was set to $T = 1,500$, while minibatch size was set to $B = 400$. Learning rate schedule for SparseVI and PSVI was $\gamma_t = 0.1 t^{-1}$. Results presented in Fig. 6 indicate that PSVI achieves superior quality to SparseVI for small coreset sizes, and is competitive to GIGA (Optimal), while the latter unrealistically uses true posterior samples to tune a weighting function required over construction.

Figure 6: Comparison of (pseudo)coreset approximate posterior quality vs coreset size for logistic regression over 10 trials.

### C.3.4 Reproducibility of Bayesian Logistic Regression experiment

In this subsection we provide additional details for reproducibility of the experimental setup for the Bayesian Logistic Regression experiment presented in Section 4.

**Posterior approximation metrics, coreset gradients and learning rates** Posterior approximation quality was estimated via computing KL divergence between Gaussian distributions fitted on coreset and full data posteriors via Laplace approximation. For both `SparseVI` and `PSVI`, gradients were estimated using samples drawn from a Laplace approximation fitted on current coreset weights and points. To optimize initial learning rates for `SparseVI` and `PSVI`, we did a grid search over $\{0.1, 1, 10\}$.

**Differential privacy loss accounting and hyperparameter selection** In the differential privacy experiment, we were not concerned with the extra privacy cost of hyperparameter optimization task. Estimation of differential privacy cost at all experiments was based on TensorFlow privacy implementation of moments accountant for the subsampled Gaussian mechanism.[1] For `DP-PSVI` we used the best learning hyperparameters found for `PSVI` on the corresponding dataset. The demonstrated range of privacy budgets was generated by decreasing the variance $\sigma$ of additive Gaussian noise and keeping the rest of hyperparameters involved in privacy accounting fixed. Regarding `DP-VI`, over our experiments we also kept subsampling ratio fixed. We based our implementation of `DP-VI` on authors code,[2] adapting noise calibration according to the adjacency relation used in Section 3.3, and the standard differential privacy definition [3]. In our experiment, we used AdaGrad optimizer [2], with learning rate $0.01$, number of iterations $2,000$, and minibatch size $200$. Gradient clipping values for `DP-VI` results presented in Fig. 4, for TRANSACTIONS, CHEMREACT100, and MUSIC datasets were tuned via grid search over $\{1, 5, 10, 50\}$. The values of gradient clipping constant giving best privacy profiles for each dataset, used in Fig. 4, were 10, 5, and 5 respectively.

### C.3.5 Additional Plots

**Evaluation of CPU time requirements** Experiments were performed on a CPU cluster node with a 2x Intel Xeon Gold 6142 and 12GB RAM. In the case of `PSVI` the computation of coreset sizes from 1 to 100 was parallelized per single size over 32 cores in total. Fig. 7 shows the posterior approximation error vs required CPU time for all coreset construction algorithms over logistic regression on the small-scale and large-scale datasets. As opposed to existing incremental coreset construction schemes, batch construction of `PSVI` reduces the dependence between coreset size and processing cost: for `SparseVI` $\Theta(M^2)$ gradient computations are required, as this method builds up a coreset one point at a time; in contrast, `PSVI` requires $\Theta(M)$ gradients since it learns all pseudodata points jointly. Although each gradient step of `PSVI` is more expensive, practically this implies a steeper decrease in approximation error over processing time compared to `SparseVI`. In the case of differentially private `PSVI`, some extra CPU requirements are added due to the subsampled Gaussian mechanism computations.

Figure 7: Comparison of (pseudo)coreset approximate posterior quality vs CPU time requirements for the logistic regression experiment of Section 4.

Figure 8: Comparison of incremental `PSVI` and existing coresets approximate posterior quality vs iterations of incremental construction (*left*) and coreset size (*right*), for the small-scale datasets logistic regression experiment. With dashed lines is displayed the posterior quality achieved by incremental `PSVI` and `SparseVI` constructions using gradients computed on data subsets of size 256.

**Incremental scheme for pseudocoreset construction** We also experimented with an *incremental scheme for pseudocoreset* construction. According to this scheme, pseudodata points are added sequentially to the pseudocoreset. Similarly to `SparseVI`, in the beginning of each coreset iteration, we initialize a new pseudodata point at the true datapoint which maximizes correlation with current residual approximation error vector. Next, we jointly optimize the most recently added pseudodata point location, along with the pseudocoreset weights vector, over a gradient descent loop. As opposed to batch construction, for large coreset sizes the incremental scheme for `PSVI` does not achieve savings in CPU time compared to `SparseVI`.

We evaluated coreset construction methods on Bayesian logistic regression. We used $M = 100$ iterations for construction, $S = 100$ Monte Carlo samples per gradient estimation, $T = 100$ iterations for optimization, and learning rate $\gamma_t \propto 0.5t^{-1}$. Coreset posterior samples over the course of construction for `SparseVI` and incremental `PSVI` were drawn from a Laplace approximation using current coreset weights and points. We implemented `SparseVI` and incremental `PSVI` via computing gradients on the full dataset, as well as using stochastic gradients on subsets of size $B = 256$ for lowering computational cost.

Results presented in Fig. 8 demonstrate that incremental `PSVI` achieves consistently the smallest posterior approximation error, offering improvement compared to `SparseVI` and even achieving better performance than `GIGA (Optimal)`. We observe that stochastic gradients implementation (dashed lines) reaches a plateau at higher values of KL compared to full gradients (solid lines), but still achieves performance comparable with `GIGA (Optimal)`.

## Footnotes

[1] https://github.com/tensorflow/privacy

[2] https://github.com/DPBayes/DPVI-code