[Reviews · NeurIPS 2020]

Review 1

Summary and Contributions: The paper extends previous work on Bayesian coresets [6] such that the selected coreset is not necessarily a subset of the observed dataset, but instead consists of any synthetic set of points that may summarize the original data more efficiently. An application to a differentially private summarization of sensitive data is proposed.

Strengths: The problem setup is relevant to the NeurIPS community, since model training and validation on large datasets can be made more efficient by reducing redundancy within the data. The paper explicitly addresses high-dimensional data through a scalable method and presents significantly better results than the selected baselines particularly on high-dimensional data. Code will be published, which allows for reproducibility and application of the proposed methods to new problems.

Weaknesses: 1. The main weakness is novelty: I consider the contribution incremental compared to [6]. In particular: 1a. The main contribution is Eq. (4), which is the analogue to [6, Eq. (6)], just with an additional variable u introduced. All definitions, proofs and derivations in Secs. 3.1 and 3.2 are basically taken over from [6] with some subtleties changed due to the new variable u. Deriving gradients in Eq. (5) and Sec. 3.2 with respect to u is convenient, however, in times of automatic differentiation not really necessary. 1b. The extension to differentially private coresets in Sec. 3.3 is simple by applying the perturbation trick from differential privacy. 2. Evaluation: A naïve implementation of differentially private coresets on the original method [6] would perturb the sensitive data in a preprocessing step. What are the advantages of your approach over this method? I would have expected that the authors compare against a naïve baseline which would be closer related to the proposed method than the other baseline.

Correctness: The proof in the appendix, Sec. B.2, was checked and seems to be correct.

Clarity: Although the paper is generally well written, it would benefit from an explicit related work section with a more detailed comparison to the methods in lines 33-36 lining out similarities and main differences. The following items in the paper remain unclear: 3. The main motivation of this method is outlined in Sec. 2.1, namely that the KL divergence scales with the dimension of the data. However, the data dimension is kept fixed during corset construction and since the KL lower bound seems to be monotonic over the coreset size given a dimensionality (see Fig. 1b), optimizing the KL lower bound would yield the best coreset size. So, I do not see a limitation here for fixed data dimensionality. Could you please clarify your argument?

Relation to Prior Work: To the best of my knowledge, the relevant related work is covered. Also see my comment above under “Clarity” w.r.t. an dedicated related work section.

Reproducibility: Yes

Additional Feedback: Minor comments: * \pi_1 in Eq. (1) is undefined. * Inconsistent notation in Definition 2: x was previously used as data point before, now it is a dataset. ------------------------------ EDIT AFTER REBUTTAL: The following of my points were not adequately addressed in the rebuttal, in particular: * Novelty: In contrast to the authors, I do not think that this idea alone is worth a paper at a tier-1 conference. * Motivation/Proposition 1 unclear: Still after the rebuttal, I do not see the point in the motivation wrt the KL growing with dimension. The authors claim that you would like to fix a threshold on the KL to optimize for a coreset and you cannot do this because the KL grows with dataset dimension. In practice, however, I guess the KL is just minimized with a given coreset budget. So my arguments in the review still apply (KL is monotonic given dimensionality). * Experiments: Evaluation of the differentially private scheme against the naive baseline is missing. This concern was not addressed in the rebuttal. I suggest to resubmit this paper after a major revision.


Review 2

Summary and Contributions: The authors propose a novel methodology that advances in the idea of using weighted coresets for scalable Bayesian inference. Instead of selecting “important” weighted samples or points of the dataset, they instead aim to find some optimal pseudo data that might be a better representation, with other new properties and the chance of increasing the number of pseudo-points more freely.

Strengths: The main strengths of the paper are: i) It is clearly written, theoretically well grounded and rigorous in the formulation all along the manuscript, i.e. I enjoyed the exponential family-like manner of presenting the problem, with the two equations before Eq. (2). Good decision. ii) The Figure 1 in the first pages is also a good point for helping readers to understand what is the problem of standard coresets and why the pseudo-data option could be interesting. iii) Experiments are well-explained and results seem convincing wrt the claims and ideas proposed. I also agree with the broader impact statement and authors were totally sincere on the undesired aspects that could appear in the future.

Weaknesses: In my opinion, the section 2 is a bit long and revisits excessively the reference number [6] that authors use to build all the formulation on. However, I understand that papers should be self-contained. My main concern comes in the decision of the authors about not including analysis or deep references to the Gaussian process framework. In Bayesian statistics, it is well-known that since 2006, the use of pseudo-data has been a crucial point for scaling up inference in stochastic processes and, more concretely, Gaussian processes. Additionally, I was surprised by the fact that the weights were maintained. It is necessary to have both w and u? would be the model also theoretically robust if removing the weights w? Another point of weakness that I detect is the shortness of the most important section in the paper (3.1 Pseudocoreset variational inference), since between L108—L130, all the critical info is included. Then, so much attention is put on the optimization part.

Correctness: I do not detect any critical mistake or error in the paper, it was well revised and both rigor and consistency are preserved in all pages. Congrats for this.

Clarity: The paper is easy to read and explanations are clear enough. The clarity in the subsection 3.3 decreases a bit wrt to the whole manuscript and I do not particularly see the interest of the differentially private scheme in the proposed methodology. That is, it is an important milestone in ML models, and it has sense in a future development of the approach, however, it could have been better if authors focused more on the analysis and consequences of the pseudo-data (that I consider novel), instead of including that paragraphs. An option instead of the privacy part, could have been to extend the model to non-Gaussian likelihoods or scenarios out of the real domain. Is the approach valid for classification as well? Would it be easy to be derived from the exponential family formulation?

Relation to Prior Work: My only concern about the prior work is the lack of connections and comments with the Gaussian process framework, it would increase a lot the quality of the paper. I say this, not due to a reason of recognition or indicate similar ideas, but because if it is included, authors from the GP literature could get new insights from the coreset works to improve their own models. Hence, maximising the impact of this paper.

Reproducibility: Yes

Additional Feedback: Minors: L215: cholesky with lower or triangular matrices? L229-L230: sub and superscripts are touching L235: M<250, I do not agree with the \approx sub-symbol Figure 2: ylabel Figures 3-4: clarify which is your method, for a better comprehension ####### [Post-rebuttal update] ############### Thanks to the authors for their answers in the rebuttal. I am now more convinced of the decisions taken by the authors and I encourage them (once again) to exploit the connection with GP models. Despite the points on the similarity with previous works as [6] and the privacy scheme, I find the idea novel and potentially useful for future literature. Consequently, I revised my score upwards.


Review 3

Summary and Contributions: In this paper, they develop a Bayesian approach for data summarization which is called Bayesian pseudocoresets where pseudocoreset construction is formulated as a variational inference problem; and variational optimization method is used to select pseudodata and its weights to generate a subset of the original dataset. That method enables high-dimensional data summarization for the efficiency of Bayesian inference while providing differential privacy for the data. The main contribution of the paper is to show the optimal pseudocoreset is able to recover the exact posterior with just one pseudodatapoint.

Strengths: I like the main contribution of the paper which is to create a subset of the data for efficiency of Bayesian inference with differential privacy guarantees. The paper is technically sound, the proposition (which is the main contribution) is proved correctly and the method can be usable in practical, real-world settings.

Weaknesses: The experiments designed are sufficient to show the practicality of the algorithm. However, in the sense of privacy, the experiments are not sufficient in my opinion. For instance, the amount of privacy protection is not clear in the experiments and the comparison results for different parameters are not given. Apart from that, the literature review in the paper is missing. Is there any other privacy protecting data summarization algorithm in the literature? Maybe it is different than coresets, but it would be beneficial to see a comparison with another privacy providing algorithm.

Correctness: I checked both the equations in the main paper and the proof and the additional experiments in the supplementary material. To the best of my knowledge, all the methods and experiments provided seem correct.

Clarity: It is mostly well-structured and well-written paper.

Relation to Prior Work: In my opinion this is the weakness of the paper. I didn't see a clear comparison of the proposed method with the methods in the literature especially for the differential privacy. I can understand the contribution and difference as a coreset algorithm but cannot see the contribution in the sense of privacy. The related work could be extended for this work.

Reproducibility: Yes

Additional Feedback:


Review 4

Summary and Contributions: Authors present a method to learn posterior distributions of models using only a weighted subset of the data or pseudo data generated to resemble the real observations for privacy preservation. Parameters are fit by minimizing the KL divergence between the coreset weighted pseudo data posterior and the true posterior via stochastic gradient optimization.

Strengths: The authors are working on an important problem in reducing the dependence of Bayesian inference on the entire data set and has appropriately discussed previous work regarding Bayesian coresets. The additional privacy preserving properties and use of pseudo data is also a very interesting addition to this method.

Weaknesses: My biggest question about this method is how the optimization technique works in the private setting. You initialize the pseudo data in the private setting by sampling data from the data generating process but if the generating process is different from the observed data then then optimizing the psuedo coreset can be very difficult. This could especially be the case in the regression setting where we would have to place a distribution on the input space. Even though your method does well in terms of KL divergence, can you discuss how well the pseudo coreset approximates the data in this setting? Moreover, your experimental results mostly show how well your method does compared to others for KL divergence but I wonder if that tells the whole story, I'd like to see how the resulting coreset posterior looks in comparison to the actual posterior especially for the real data examples.The CPU wall time results in the Supplement seem like the comparing methods weren't run to convergence, judging by the fact that the KL doesn't flatten out, it's hard to argue that your method really performs better there especially since it's more computationally expensive to run compared to other methods.

Correctness: The method overall seems correct.

Clarity: The overall presentation of the paper is clear.

Relation to Prior Work: Yes, the paper has addressed a lot of previous well known work in Bayesian coresets, particularly with the Broderick papers in coresets.

Reproducibility: Yes

Additional Feedback: Author Feedback Update: Thank you for your rebuttal. I am satisfied with the authors' responses and I vote to accept the paper.

[Author Response · NeurIPS 2020]

We thank reviewers for their constructive feedback on our work. We are happy to see that the problem of data summa-
rization for scalable and privacy-preserving Bayesian inference in high-dimensions is recognized as important for the
community, and our approach was found technically sound and usable in real-world applications.
**R2 (novelty):** Although pseudodata-based sparsifications for VI are not new in ML, this idea is novel and nontrivial in
the context of summarization. It also provides three key benefits that are specific to this setting, namely: it enables (1)
summarization in high-dimensions, (2) private release of summarizations, and (3) batch construction (reducing complex-
ity). Further, our derivations take advantage of the particular form of our objective/gradients for efficient computation.
**R3 (connections with GP literature):** Sparse methods in GPs have been indeed inspiring for delevoping PSVI. Simi-
larly to (Titsias, 09) and in contrast to (Seeger et al., 03; Snelson and Ghahramani, 06), (1) we learn pseudopoints via
continuous optimization on the KL between the exact and approximate posterior, avoiding overfitting by construction,
and (2) do not modify the model prior, but introduce pseudopoints as variational parameters. Note that our method is
applicable in both supervised and unsupervised learning settings. We will expand on these connections in our revision.
**R3 (applications to broader likelihood functions):** Our method is agnostic to the particular form of data likelihood
functions and can be readily applied to classification problems (see e.g. the Bayesian logistic regression experiment
in Section 4 and Supplement C.3.1). We will emphasize that PSVI maintains the unifying sparse exponential family
interpretation for any statistical model it is applied to, with pseudopoints weights and likelihood terms corresponding
respectively to the natural parameters and sufficient statistics for the family of pseudocoreset posterior approximations.
**R2 (clarity of Proposition 1):** In Gaussian mean inference, under the standard coresets formulation, for fixed dataset
size and data dimensionality, the minimum required coreset size to reduce KL below a given threshold is bounded per
Proposition 1. This bound depends on the dimension and increases when summarising a dataset of larger dimensionality,
as empirically demonstrated in the difference between the KL plots of baseline methods in 200 and 500 dimen-
sions (Fig. 2(a) and (b) of the paper)—implying impractical summary sizes for good KL. Pseudocoresets are not con-
strained by this bound and can achieve arbitrary KL reduction by a single pseudopoint, regardless of data dimensionality.
**R5 (pseudodata and posterior quality):** Learned pseudodata are explicitly optimized to approximate (in the KL
sense) the exact posterior for a given statistical model, forming "approximate sufficient statistics" of the full data.
Ongoing experiments showed us that pseudocoreset posteriors can be successfully
applied in predictive analysis offering improvements in test accuracy/rmse. Though
Hilbert coresets and uniform sampling might eventually achieve higher KL reduction
for (often prohibitively) large coreset sizes, we are primarily interested in small
coresets, where PSVI is outperfoming baselines in the tradeoff of KL reduction,
coreset size and CPU time (required for both summary construction *and* subsequent
inference); in contrast, Hilbert coresets are fundamentally constrained in this regime
both due to data dimensionality (as is SparseVI as well), and information-geometric
limitations (see (Campbell & Beronov, 19) and plot shown on the right).

**R3 (pseudodata weights):** The varational parameters size in PSVI is dominated by pseudopoints in high-dimensions.
Weights seem to be a natural ingredient for data summarization, that can account for coreset points multiplicity, hence
enabling more expressive sparse posteriors, without having a significant bearing on the computational cost and the
robustness of optimization. Importantly weights can differentiate posterior approximations among datasets of different
size. For example, removing the variational parameters $w$ in the Gaussian mean inference experiment (Section 4), won't
allow correctly adjusting the covariance of the pseudocoreset posterior, which is not a function of pseudopoints location.
**R2,3,5 (private scheme):** A major desideratum in Bayesian coresets is maximising the automation of inference. Using
the subsampled Gaussian mechanism is a decisive step towards pursuing this goal in DP extensions of coresets: our
privatisation method removes requirements on computing sensitivies for noise calibration, enables adaptive clipping of
gradients guided by private statistics on pseudopoints potentials, and gives tight estimates of the accumulated privacy
cost via moments accounting—the latter allows many gradient steps under DP leading to good convergence in KL in
practice, even when pseudodata are initialised from an uninformed prior, potentially far from true observations (Section
4). On the other hand, privatising via noise addition in the first place requires strong public knowledge/assumptions
on the (typically infinite) data likelihood sensitivities. Moreover, exponential mechanism based private selection for
incremental schemes of summarization would not allow tight composition of privacy over a large number of iterations.
**R4 (privacy evaluation and related work):** We kept $\delta$ parameter fixed to $1/N$ over all experiments on private
inference, as this allows reasonable relaxations of pure DP quarantees. Fig. 4 of the paper presents the achieved
posterior approximation quality over a range of values for the $\varepsilon$ parameter for both our method and the baseline, profiling
methods behavior over the regime of strong and weak privacy guarantees. DP schemes for coresets applicable in
computational geometry already exist (Feldman et al., 09; 17), whilst the idea of releasing private dataset compressions
has been also pursued in kernel methods (Balog et al., 18), sparse regression (Zhou et al., 07), and compressive
learning (Schellekens et al.19); however, none of these approaches is directly applicable to summarising for general-
purpose Bayesian inference, which led us to the decision of comparing against a standard private VI method.
**R2,3 (clarity of presentation, minor comments):** We will address all typos, fix inconsistent notation, adapt sections
length and clarity according to your suggestions, and expand on the noted references. Thank you for pointing these out.

[Meta-Review · NeurIPS 2020]

Although the reviewers consider the paper somewhat incremental, after the author response and discussion, a clear majority supports acceptance as the method appears useful and with potential to have significant impact. The reviewers note a number of weaknesses in the original submission (e.g. lack of proper reporting of privacy levels, lack of baselines for DP methods). I would strongly encourage the authors to address all of these in the final version, including ones that were omitted in the rebuttal (presumably because of lack of space).